



# Level0-to-Level1B processor for MethaneAIR

Eamon K. Conway[1,6], Amir H. Souri[1], Joshua Benmergui[2], Kang Sun[3,4], Xiong Liu[1], Carly Staebell[3], Christopher Chan Miller[1], Jonathan Franklin[2], Jenna Samra[1], Jonas Wilzewski[1], Sebastien Roche[1], Bingkun Luo[1], Apisada Chulakadabba[2], Maryann Sargent[2], Jacob Hohl[1], Bruce Daube[1], Iouli Gordon[1], Kelly Chance[1], and Steven Wofsy[2,5]

[1]Center for Astrophysics | Harvard and Smithsonian, Atomic and Molecular Physics Division, Cambridge, MA, USA. 02138
[2]Harvard John A. Paulson School of Engineering and Applied Sciences, Harvard University, Cambridge, MA, USA
[3]Department of Civil, Structural and Environmental Engineering, University at Buffalo, Buffalo, NY, USA
[4]Research and Education in Energy, Environment and Water Institute, University at Buffalo, Buffalo, NY, USA
[5]Department of Earth and Planetary Sciences, Harvard University, Cambridge, MA, USA
[6]Kostas Research Institute, Northeastern University, Burlington, MA, USA

**Correspondence:** Eamon K. Conway (e.conway@kri.neu.edu)

**Abstract.**

This work presents the development of the MethaneAIR L0–L1B processor, which converts raw L0 data to calibrated and georeferenced L1B data. MethaneAIR is the airborne simulator for MethaneSAT, a new satellite under development by MethaneSAT LLC, a subsidiary of the EDF. MethaneSAT's goals are to precisely map over 80% of the production sources

5 of methane emissions from oil and gas fields across the globe to an accuracy of 2–4 ppb on 2 $km^2$ scale. Efficient algorithms have been developed to perform dark corrections, estimate the noise, radiometrically calibrate data, and correct stray-light. A forward model integrated into the L0–L1B processor is demonstrated to retrieve wavelength shifts during flight accurately. It is also shown to characterize the ISRF changes occurring at each sampled spatial footprint. We demonstrate fast and accurate orthorectification of MethaneAIR data in a three-step process: i) initial orthorectification of all observations using aircraft

10 avionics, a simple camera model, and a medium resolution digital elevation map, followed by ii) registration of oxygen ($O_2$) channel grey-scale images to reference Multispectral Instrument (MSI) band 11 imagery via Accelerated-KAZE (A-KAZE) feature extraction and linear transformation, and similar co-registration of methane ($CH_4$) channel grey-scale images to the registered $O_2$ channel images, and finally iii) optimization of the aircraft position and attitude to the registered imagery and calculation of viewing geometry. This co-registration technique accurately orthorectifies each channel to the referenced MSI

15 imagery. However, in the pixel domain, radiance data for each channel are offset by almost 150-200 across–track pixels (rows) and need to be aligned for the full physics or proxy retrievals where both channels are simultaneously used. We leveraged our orthorectification tool to identify tie points with similar geographic locations in both $CH_4$ and $O_2$ images in order to produce shift parameters in the across-track and along-track dimensions. These algorithms described in this article will be implemented into the MethaneSAT L0–L1B processor.



# 1 Introduction

Methane ($CH_4$) is a potent greenhouse gas that, despite having an atmospheric lifetime of approximately twelve years, almost one eight to that of carbon dioxide, it has approximately 26 times the global warming potential (GWP) of $CO_2$ over a 100 years time horizon. Over the first 20 years, methane has a GWP approximately 70-80 times that of $CO_2$ (Forster et al., 2007). Analysis carried out by Etheridge et al. (1998) on ice-core samples extracted in the Antarctic peninsula showed atmospheric methane concentrations have rapidly increased since the 1700s. These increases are predominantly anthropogenic, driven by industrialization and agricultural demand (Hmiel et al., 2020). In 2017, atmospheric levels of $CH_4$ averaged 1850 ppb, triple the eighteenth-century limits of approximately 650 ppb (Nisbet et al., 2019). Despite knowledge of its large atmospheric warming potential, except for a 'stabilization' period between 2000 and 2007, methane concentrations continue to increase (Turner et al., 2019). Preliminary results from the National Oceanic and Atmospheric Administration (NOAA) suggest that 2020 may have seen the largest increase in atmospheric methane concentrations since 1983 (NOAA, 2021).

Methane monitoring instruments such as TROPOMI (Veefkind et al., 2012) can detect emission plumes to a 12 ppb precision on a $7 \times 5$ km$^2$ scale (Pandey et al., 2019; Zhang et al., 2020) at nadir. TROPOMI's 2600 km swath provides daily global coverage, which greatly increases our knowledge of emission hot-spots on the Earth, but the coarse resolution makes it challenging to quantify and detect small-scale sources. MethaneSAT (MethaneSAT LLC) will be launched sometime in the first quarter of 2024, and the mission aims to precisely map over 80% of the production sources of methane emissions from oil and gas fields across the globe to an accuracy of 2–4 ppb on 2 km$^2$ scale, with a native spatial resolution of 400 m $\times$ 100 m (along and across track respectively) at nadir, supporting the goal of reducing methane emissions from the oil and gas sector by 45% by the end of 2025. It will have two spectrometers on board, an oxygen ($O_2$) sensor, and a methane ($CH_4$) sensor, both of which will operate in the shortwave infrared region. The $CH_4$ sensor will observe between 1598 nm and 1683 nm, which encompass the $CO_2$ absorption features at approximately 1610 nm and the $CH_4$ features at approximately 1650 nm. The $O_2$ sensor will observe between 1249 and 1305 nm, encompassing the 'singlet-delta' $O_2$ $a^1\Delta_g$ - $X^3\Sigma_g^-$ band at approximately 1270 nm. The swath range will encompass approximately 200 km at an altitude of approximately 526 km, with $\approx$ 2000 pixels across the track, enabling the precise detection and identification of point source emissions.

It has been more common to use the oxygen A-band at 762 nm to determine the optical path length along the line of sight, with the SCanning Imaging Absorption spectroMeter for Atmospheric CHartographY (SCIAMACHY), Greenhouse Gas Observing Satellite (GOSAT), TROPOspheric Monitoring Instrument (TROPOMI) and Orbiting Carbon Observatory-2 (OCO-2) all observing this band. Many experimental measurements of this band have hence been carried out to improve the quality of spectroscopic information (Drouin et al., 2017, 2013), which include the observation of electric quadrupole transitions (Miller and Wunch, 2012). However, recent efforts using cavity ring down spectrometers have significantly improved the spectroscopic information of the singlet-delta band. Konefał et al. (2020); Tran et al. (2020) have shown that prior transition intensities were incorrect by between 1-3%, a non-negligible and important improvement for the MethaneSAT mission. MethaneSAT will also need to model the airglow in this band, which can pose a challenge. However, despite this, recent advances in airglow





simulations using SCIAMACHY data have used this band and excellent results are attainable (Sun et al., 2018; Bertaux et al., 2020), which show that this band can be used for satellite remote sensing of greenhouse gasses.

Airborne simulators provide an opportunity to develop and validate science algorithms prior to obtaining observation data from a forthcoming space-based instrument. Considering the NASA TEMPO (Tropospheric Emissions Monitoring of Pollution) (Zoogman et al., 2017) geostationary satellite mission that will operate in the visible and ultraviolet spectral regions, Geostationary Trace gas and Aerosol Sensor Optimization (Geo-TASO) (Leitch et al., 2014; Nowlan et al., 2016), Airborne Compact Atmospheric Mapper (ACAM) (Kowalewski and Janz, 2009; Lamsal et al., 2017; Nowlan et al., 2018) and GEOsta-

tionary Coastal and Air Pollution Events (GEO-CAPE) Airborne Simulator (GCAS) are examples of airborne simulators that have provided the resources to design and enhance the retrieval algorithms. In the SWIR, AVIRIS (Thompson et al., 2016), MAMAP (Gerilowski et al., 2011), GHOST (Humpage et al., 2018) and JAXA's airborne instrument (Kuze et al., 2020), have been utilized to retrieve methane emissions. The airborne simulator for MethaneSAT is appropriately named MethaneAIR and it includes two spectrometers scanning almost the same SWIR regions as MethaneSAT. Ten research flights have been

completed in different parts of the US, with the final flight consisting of airglow observations. These research flights provided high-quality observation data to develop a fully automated MethaneAIR L0–L1B processor. The data sets have proved invaluable in developing science algorithms for MethaneSAT. Four additional research flights were performed in the Fall of 2022, but will not be considered in this paper.

**Table 1.** Shown below are the details for each MethaneAIR deployment, including destination and approximate length of observation

| Flight | Date | Destination | Time (hrs) |
|--------|------|-------------|------------|
| RF01 | 20191108 | CO Frontal Range | 2.3 |
| RF02 | 20191112 | CO Frontal Range | 3.6 |
| RF03 | 20210728 | CO Frontal Range | 3.2 |
| RF04 | 20210730 | Permian | 5.5 |
| RF05 | 20210803 | Permian | 6.4 |
| RF06 | 20210806 | Permian | 6.2 |
| RF07 | 20210809 | Permian | 6.8 |
| RF08 | 20210811 | Salt Lake City & Uinta Basin | 6.1 |
| RF09 | 20210813 | Bakken | 7.3 |
| RF10 | 20210816 | Airglow | 4.7 |

It is not uncommon for instrument properties such as wavelength registration and instrument line shape to slowly deviate away from the measurements recorded on the ground (Sun et al., 2017a, b; Voors et al., 2006; Cai et al., 2022). Such variations can be a result of the vibration during the initial launch of the instrument or from the differences between the environmental conditions encountered in the air (or space) versus the laboratory setup on the ground. It is therefore quite useful to routinely perform in-flight calibration measurements to monitor for any possible changes. For OCO-2, Sun et al. (2017b) analyzed the observed instrument line shape (ILS) and wavelength registration and compared these to the ground-based calibration measurements. Several line shape models were tested, super-Gaussian (Beirle et al., 2017), Gaussian, stretch/squeeze of the pre-flight ILS, and a hybrid mix of Gaussian functions (Liu et al., 2015). It was found that the flat-top nature of the

OCO-2 ILS made the Gaussian function unsuitable, however, reasonable results were obtained with the other models. While the TROPOMI instrument has dedicated onboard calibration light sources (van Hees et al., 2018; Van Kempen et al., 2019), the



laboratory-measured wavelength calibration parameters (Kleipool et al., 2018) are not monitored in the L0–L1B processing, but rather during trace gas retrievals.

The MethaneAIR spectral calibration measurements have already been described in great detail by Staebell et al. (2021):
Wavelength, radiometric and straylight properties, together with the instrument response function (ISRF), were derived for each sensor. The measured data have been implemented into the MethaneAIR L0–L1B processor. In this article, the development of the L0–L1B processor will be described in detail.

## 2   The MethaneAIR Instrument

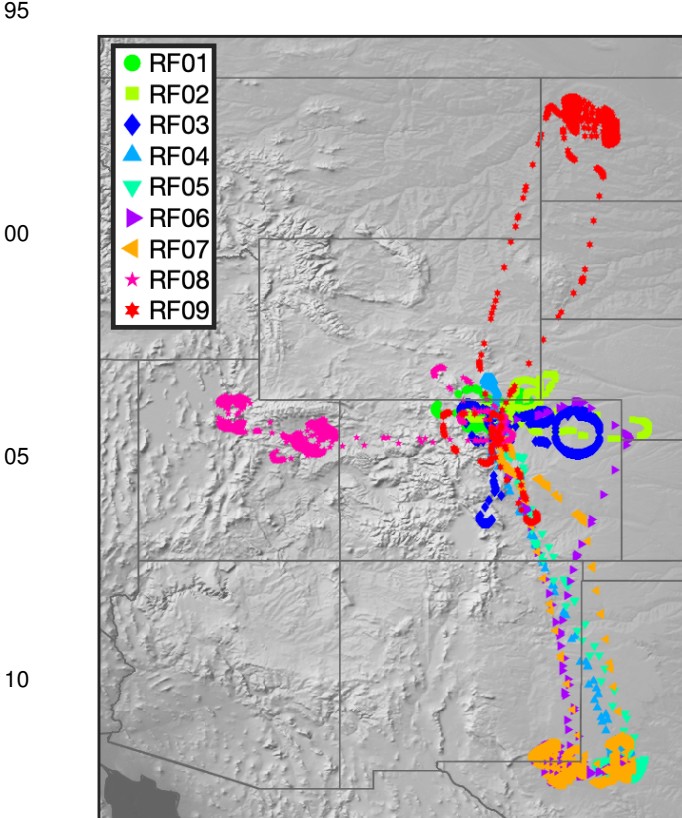

**Figure 1.** Map of the MethaneAIR research flights.

The MethaneAIR instrument closely echos the design of MethaneSAT. Two spectrometers, one measuring methane spectra and the other oxygen spectra, were integrated side-by-side onto the National Science Foundation (NSF) Gulf-stream V (GV) aircraft that is controlled by the National Center for Atmospheric Research (NCAR) (NCAR - Earth Observing Laboratory, 2005). Each of these detectors possesses 1024 spectral pixels (columns) and 1280 spatial pixels (rows). The bandpass for the $CH_4$ sensor covers 1592–1680 nm, while the $O_2$ sensor spans 1236–1319 nm. Of the 1280 rows, only indices 135–997 and 308–1170 are illuminated in each of the respective $CH_4$ and $O_2$ sensors. The GV was equipped with an anti-reflection coated window and for the wavelengths, we are interested in, 1236–1680 nm, the aircraft window transmittance changes smoothly from 99.7% to 98.1%. The native spatial resolution of MethaneAIR observations is approximately 25 m (along-track) x 5 m (across-track) for a typical flight altitude of 9 km. More details of the MethaneAIR instrument are described by Staebell et al. (2021).

### 2.1   Research Flight Details

The dates, destinations, and observation times of the ten research flights we will consider in this paper are described in Table 1 and Figure 1. In total, approximately 50 hours of observation data were recorded for MethaneAIR. The first two flights, RF01 and RF02, which occurred in late 2019, were





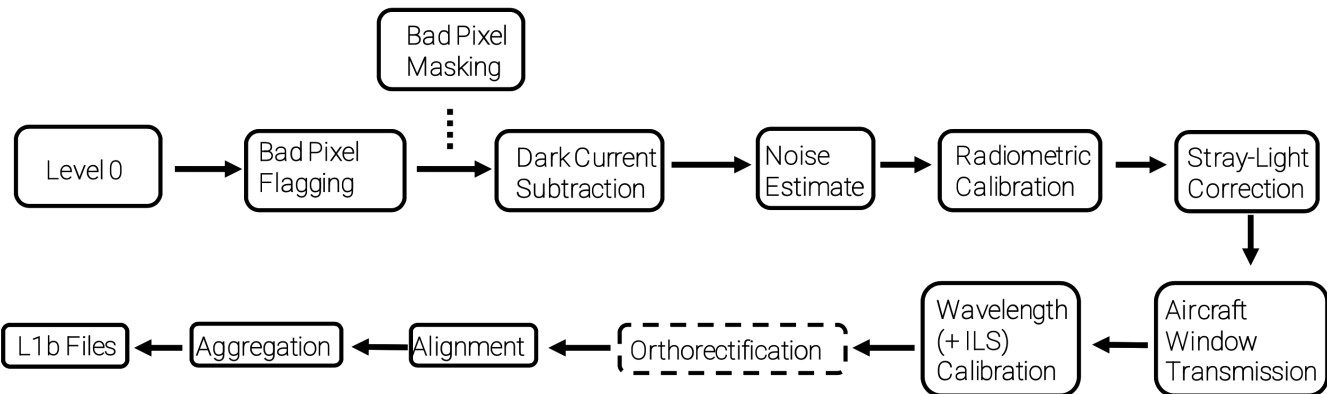

**Figure 2.** Flowchart representation of the processes within the MethaneAIR L0–L1B processor. The orthorectification process is in a dashed box to show that it is written in R code, and imported into Python.

engineering flights, whose purpose was to ensure that the instrument was operating as expected. Over these two flights, 5.8

hours of observation data were collected in the Colorado Frontal Range. More flights were planned to occur at this time, however, flights ended due to GV maintenance issues and were rescheduled for a later date. Due to the COVID-19 outbreak, these rescheduled MethaneAIR flights were further delayed until the summer of 2021. Hence, these engineering flights proved to be critical contributors to the development of the science algorithms of both MethaneAIR and MethaneSAT.

RF03 was the first of these new flights and it covered a similar region as RF02. Its purpose was to validate both the instru-

ment performance and science algorithms via comparisons to RF02, prior to mapping out targets encompassing oil and gas infrastructure. For RF04–RF07, MethaneAIR mapped out regions of the Permian Basin over the course of 25 hours (including transit time from Colorado). During RF04 and RF05 research flights, numerous passes over controlled methane release experiments also occurred. These releases provide valuable ground truth to evaluate our methane retrieval and emission estimate algorithms

Almost 6.1 hours of observation data were collected during RF08, which targeted Salt Lake City and the Uinta basin. For RF09, the Bakken oil and gas region that encompasses North Dakota was the designated target. For visual context, a map detailing the flight paths of RF01–RF09 is shown in Figure 1. RF10 was dedicated to investigating airglow emission, a key challenge for MethaneSAT's observations in the $O_2$ singlet-delta band. To measure airglow emission, the MethaneAIR payload was rotated 180 degrees so that the $O_2$ sensor could look vertically out of the upper port aboard the NCAR/GV aircraft.

## 3   L0–L1B Development

Figure 2 displays the processes of the L0–1B algorithm. Each processing step is described below from Sect. 3.1 through Sect. 3.8 and the process is fully automated. For 30 seconds of L0 data, the CPU time is approximately 90 minutes. Most of the run time is dedicated to the orthorectification procedures, primarily the final optimization procedure. The orthorectification





procedure shown in Figure 2 is within a dashed enclosure as the process has two separate steps to it. The final one is a
refinement procedure and is technically optional, but provides more accurate geospatial information onto the pixels. The L0-
L1B processes have also been configured to operate in a cloud environment, and for future flights of MethaneAIR, processing
will be done in the cloud.

## 3.1  Dark Current and Bad Pixel Flagging

During each of the research flights, dark collects were obtained at different instances during the flights and each was used to
filter bad pixels according to their standard deviation from the mean. By averaging together all frames in each of the 30-second
collects in the along-across direction, a mean dark current was derived and a 3-$\sigma$ limit was used to identify the bad pixels, i.e.
those pixels whose signals deviate from the mean by more than 3-$\sigma$. Applying this limit to each sensor, an average of 1617 and
467 bad pixels were identified in each of the $O_2$ and $CH_4$ sensors, respectively. Staebell et al. (2021) used a similar approach
to flag bad pixels, using a 3-sigma deviation limit from the linear fit of the gain of each sensor.

These bad pixels were flagged/masked in all subsequent processes. Next, the mean dark current is subtracted from each
illuminated collect to yield dark corrected frames.

## 3.2  Noise Estimate

The XCH4 retrieval follows an optimal estimation framework adjusting the concentration of several absorbing trace gases
and atmospheric states (i.e., temperature and pressure) to best match the observed radiance. The noise model determines
the uncertainty of the observed radiance populating the covariance matrix of observations in the retrieval algorithm. Good
knowledge of the noise level is critical in reducing the contributions of observational errors to the retrieved XCH4. Using the
dark collects, read noise ($N_r$) is calculated for each pixel by considering the standard deviation of the signal across the number
of frames collected. Combining the mean dark current ($D$) in units of digital number (DN), measured raw signal ($S$) in DN,
gain ($g$) in units of electrons/DN, the number of dark frames ($N_f$) with the electronic offset ($\epsilon$) in DN, the noise (in DN)
of a particular pixel is calculated via Eqn. 1. The CCD gain, defined as the slope of the variance to the median signal, was
determined to be 4.6. These values will be converted to radiance and stored in the L1B files.

$$\frac{\sqrt{(S-\epsilon)*g + (\frac{D-\epsilon}{N_f})*g + (N_r*g)^2}}{g} \tag{1}$$

## 3.3  Radiometric Calibration

Radiometric calibration converts raw sensor data in units of DN to radiance units. The radiometric calibration is based on labo-
ratory measurements using an integrating sphere with a calibrated broad band light source. A fifth-order polynomial is derived
and subsequently applied to each illuminated pixel, whose coefficients originate (for RF01 and RF02) from the radiometric
calibration measurements to convert DN to photons $s^{-1}$ $cm^{-2}$ $nm^{-1}$ $sr^{-1}$. After the first two research flights were concluded,
newer radiometric calibration files were created after an extensive calibration effort, using the same approach as Staebell et al.
(2021). These newer coefficients were utilized in the processing of RF03–RF10. In situations where we have only a small



number of dark current collects for a research flight (less than two for RF01 and RF02), it can increase the possibility of random telegraph signals (RTS) going unchecked if a second bad pixel filtering procedure is not implemented on the illuminated scenes. Hence, to screen for extreme outliers, any pixel with radiance over $10^{15}$ photons s$^{-1}$ cm$^{-2}$ nm$^{-1}$ sr$^{-1}$ or below $10^{10}$ photons s$^{-1}$ cm$^{-2}$ nm$^{-1}$ sr$^{-1}$ is also flagged as a bad pixel during processing. These limits were chosen as they are outside of the expected operating range of the sensors. In addition to the bad pixel map, two pixels were found to be problematic in the

CH$_4$ channel of the first two research flights and were hence flagged. The radiometric calibration coefficients were identified to be the source of this error and were masked. None were captured in RF03–RF10, which used new calibration measurements.

### 3.4 Stray Light Correction

Stray light is undesirable electronmagnetic radiation that reaches the sensor and interferes with performance and can occur for various reasons. The MethaneAIR sensors were designed to minimize the presence of stray light. Ground-calibration efforts
confirmed that less than 2% of the total light recorded by the sensors is traceable to stray light effects. A full description of the characterization of MethaneAIR stray-light kernels is described by Staebell et al. (2021), but we briefly describe it here. The stray light correction algorithm of Staebell et al. (2021), is based on the method used to correct the stray light in the TROPOMI instrument (Tol et al., 2018). The central region of 15 spectral × 11 spatial pixels of the stray light kernel is defined as in the band, and the kernel values in the central region are set to zero, so that and the stray light correction is applied to pixels outside
of this central area. This modified kernel is referred to as the far-field kernel and was utilized in the stray light correction algorithm, where the light is iteratively deconvolved and redistributed.

### 3.5 Wavelength and ILS Calibration

Successful trace gas retrievals rely on accurate knowledge of the observed wavelength and the instrument line shape (ILS). Laboratory calibration efforts were carried out to measure these values pre-flight (Staebell et al., 2021); however, it is relatively
common for the observed wavelength to deviate from the laboratory-calibrated measurements during operation. It is, therefore, necessary to supplement this process with a wavelength-fitting routine to adjust the coefficients during flight. Also, unlike TROPOMI, the MethaneAIR and MethaneSAT instrument configurations are not designed to allow us to accurately monitor the instrument line shape (ILS) during operation with calibrated on-board lasers. We therefore added a forward model into the L0–L1 processor that can reliably detect abrupt changes in the ILS during flight, however, this process is optional in the L0-
L1B processor for MethaneAIR because precise inverse modelling occurs during the L1-L2 processing step where wavelength and ILS fitting parameters are retrieved. The forward model we consider begins with a high resolution ($HR$) (0.001 nm) solar spectrum $F_{solar}^{HR}(\lambda)$ (Coddington et al., 2021), where $\lambda$. The parameterization of $\lambda$ with respect to spectral pixel index is assumed throughout.

$$I(\lambda) = \int \underbrace{(e^{-\tau(\lambda+\lambda'+\delta\lambda)})^{HR}}_{A} \underbrace{F_{solar}(\lambda+\lambda'+\delta\lambda)^{HR}}_{B} \underbrace{L\left(\sum_i p_i(\lambda+\lambda'+\delta\lambda)\right)}_{C} d\lambda' \times \underbrace{\left(\sum_{k=0} Al_k \times (\lambda+\delta\lambda)^k\right)}_{D} + \underbrace{\sum_{j=0} B_j \times (\lambda+\delta\lambda)^j}_{E}$$

(2)





The first term in our forward model above, $A$, is the Lambert-Beer term that accounts for light attenuation in the atmosphere due to the presence of gases, which we will consider to be made up of water, methane, and oxygen. The next, $B$, is a high-resolution solar spectrum. $C$ described the slit function or ILS, which is derived from parameters $p_i$. $D$ is a polynomial scaling term and finally, $E$ is a baseline correction term.

For $CH_4$, cross sections are calculated from the line-by-line parameters in the HITRAN2020 (Gordon et al., 2021) database
using the Voigt profile on a 0.001 nm grid for a variety of temperatures and pressures that cover the standard U.S. atmosphere ranges. For $O_2$-$O_2$ collision-induced absorption (CIA), we attempted to use several sources of data in our model, each of which is theoretically, experimentally or empirically derived. For theoretical CIA, we attempted to utilize the calculations from Karman et al. (2018), which we obtained from HITRANonline (Karman et al., 2019), for the experimental source, we tried to use the measurements from Maté et al. (1999), while we also considered the empirical TCCON CIA. For the absorbing species,
the cross sections are weighted according to the standard US atmosphere with temperature ($T$) and pressure ($P$) at atmospheric layer $i$:

$$\sigma(\lambda) = \frac{\sum_i^{max} \sigma(\lambda, T_i, P_i) \times SCD_{geo}(T_i, P_i)}{\sum_i^{max} SCD_{geo}(T_i, P_i)} \tag{3}$$

where $SCD_{geo}$ refers to the geometric slant column density.

The high-resolution data need to be convolved with respect to the measured ILS, which was probed at a number of wave-
lengths for both channels. For the methane channel, we have nine ISRF measurements at 1593 nm, 1600 nm, 1610 nm, 1620 nm, 1630 nm, 1640 nm, 1650 nm, 1660 nm, and 1670 nm, while for the oxygen channel we have ten measurements at 1254 nm, 1261 nm, 1268 nm, 1275 nm, 1282 nm, 1289 nm, 1296 nm, 1303 nm, 1310 nm and 1317 nm. The ILS at the unmeasured wavelengths was determined by interpolating the measured data points. The forward model has the option of fitting each ISRF using either a squeeze/stretch parameter applied to the LUT values, or a normalized Gaussian or super-Gaussian function. The
parameters describing the slit function $L$ are denoted as $p_i(\lambda)$ where the index runs over the number of possible parameters needed in each function for each ISRF: the Gaussian model has only one, squeeze/stretch has one while the super-Gaussian possesses two variables.

In Figure 3, the results of a single fit to observed $O_2$ and $CH_4$ spectra taken during RF01(a,c) and RF02(b,d) is demonstrated. In each case, 100 along track frames have been aggregated together to enhance the signal-to-noise ratio. In all cases, a third-
order polynomial was used to fit a baseline, while a constant shift parameter was found to provide the best results.

In Figures 3a and 3b, example fits in the $CH_4$ channel are presented for the engineering flights, RF01 and RF02. In both instances, modeling the ILS with a Gaussian, super-Gaussian, or squeeze/stretch term is shown to provide lower residuals when compared to a simulated spectrum using the initial look-up table of ISRFs, derived by Staebell et al. (2021). It is apparent that there are significantly lower residuals in RF01 than in RF02, and this can be explained by the cabin temperature not being
controlled during RF02, and that the signal is higher in RF01. Analyzing both of these figures, it is quite clear that the measured ISRF FWHMs are too narrow. For the $CH_4$ band at 1630–1650 nm, the forward model can simulate the dominant observations with an average residual of 1–2% with either a Gaussian, super-Gaussian or squeeze model while using the LUT ISRFs yields

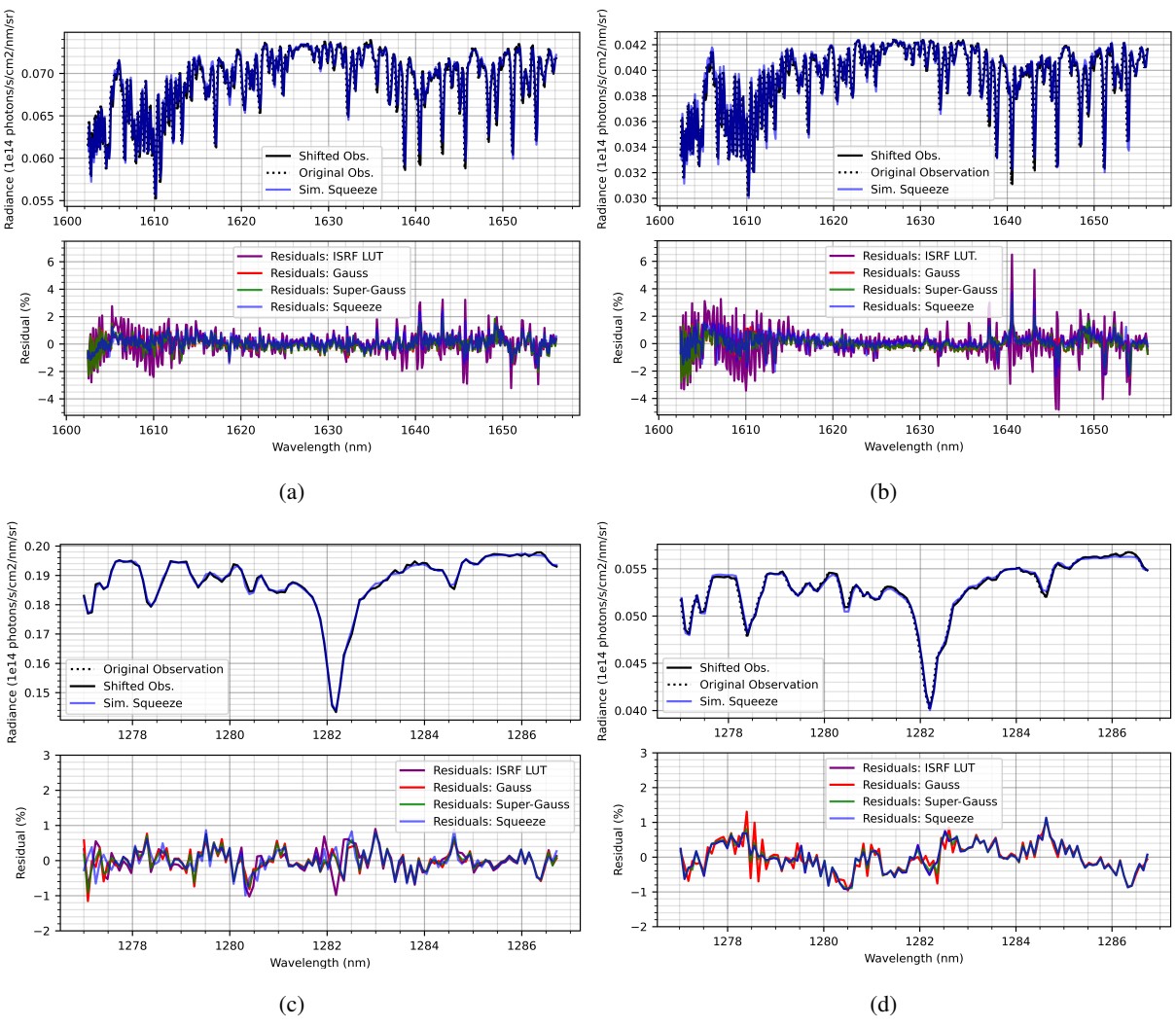

**Figure 3.** Example spectrum fits to RF01 and RF02 for $CH_4$ (a,b) and $O_2$ (c,d) channels respectively, using various ISRF models: laboratory measured values, Gaussian, super-Gaussian, and squeeze of ISRF LUT. Only the squeeze model simulation spectra are shown in the upper panels to enhance the visibility of the observed wavelength shift.



residuals closer to 3-6% in the strong absorption features. At shorter wavelengths near 1605 nm, the residuals for the $CO_2$ band are within 1% with each of a Gaussian, super-Gaussian, or squeeze ISRF model.

For RF02, the wavelength was observed to shift between 0.12–0.15 nm throughout the flight, a large portion of which is likely due to the temperature variation on board. The shift is also cross-track dependent. For RF03, the wavelength shift was significantly larger, averaging 0.2 nm. The wavelength shifts for the remaining flights were significantly less, averaging approximately 0.01–0.05 nm. The derived wavelength shifts in all cases were rather insensitive to the choice of slit function used in the forward model.

For the oxygen singlet-delta band, simulating the strong Q branch was shown to be very problematic. Regardless of our choice of reference CIA data, or baseline polynomial, reasonable fitting results were not achievable using data from either research flight. We believe the difficulty is due to a combination of the CIA not being modelled correctly, large variations in the ISRF, and the simplicity of the forward model. Hence, we instead consider fitting the solar line at approximately 1282 nm and remove CIA as an absorbing species in the forward model. For each ISRF model considered, including the ISRF LUT

values, residuals averaging less than 1% are attainable. This is true for all research flights. Example fits are shown for RF01 in Figure 3c and RF02, Figure 3d. For all flights, wavelength shifts in this channel are reasonably smaller than their $CH_4$ counterparts, averaging less than 0.05 nm.

The observed ISRF FWHMs for all research flights deviate from the laboratory-measured values from Staebell et al. (2021). As an example, we consider RF02, which was a flight where the variation is considerably large, evident from Figures 3a and

3b. Lower residuals are attainable for observations recorded in other research flights, likely a consequence of cabin temperature being controlled and monitored during flight, however, for RF02, the temperature was much less stable (not recorded). While this is clearly not an ideal situation for performing science retrievals, it does present the perfect opportunity to develop robust science algorithms.

The temperature variation in the aircraft cabin has likely resulted in the ILS changing throughout the RF02 flight. In Figure 4,

the LUT ISRF data are fitted with a Gaussian line-shape and the resulting FWHMs are compared to the retrieved Gaussian FWHMs deduced from RF02 data at wavelengths: 1610 nm, 1620 nm, 1630 nm, 1640 nm, and 1650 nm. The results shown are for a thirty-second granule captured when the aircraft was at 2.26 km and the granule consists of over 100 frames which were aggregated along–track. Dramatic changes in each ISRF are evident and differences of up to approximately 30% are observed. The pattern seems to be almost reversed with respect to the across–track dimension, evidence of large environmental changes

surrounding the instruments on board. Despite these large variations, relatively low residuals are obtained during wavelength calibration.

## 3.6 Orthorectification

Orthorectification is the process of mapping camera projections onto the Earth's surface to accurately determine the geolocation of the measurements. In order to pinpoint the exact location of methane emission sources, retrievals of column-average mole

fractions of methane ($XCH_4$) from MethaneAIR data have three major sensitivities to orthorectification accuracy:





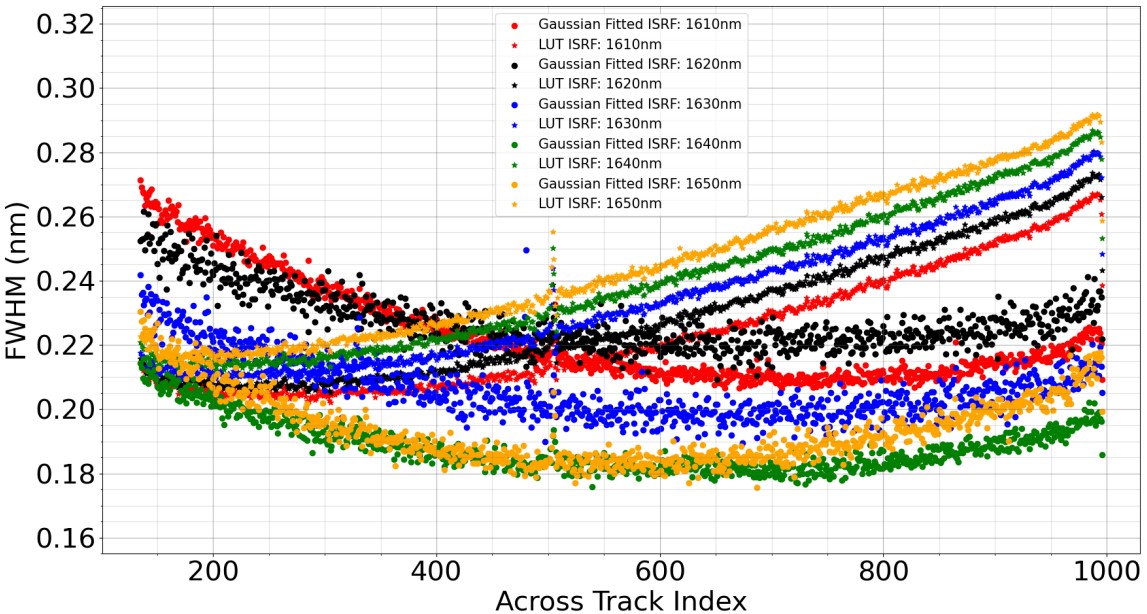

**Figure 4.** Retrieved ISRF FWHM using a Gaussian model obtained from a 30-second granule in RF02 compared to the laboratory ISRF that was fit to a Gaussian for equal comparison.

1. The MethaneAIR instrument is composed of separate cameras for $O_2$ and $CH_4$ bands that must be spatially registered and aligned,

2. The retrieval algorithms require information about surface properties that vary widely over small spatial scales, and

3. The retrieval algorithms require information about the viewing geometry of the measurements.

Note that the sensitivity of the retrieval algorithms to the viewing geometry is not only a sensitivity to the camera projections onto the Earth's surface but also a sensitivity to the position and attitude of the camera itself. We apply an orthorectification procedure that makes a first guess using the aircraft avionics, then refines the mapping to the Earth's surface with an image registration algorithm, and finally refines the position and attitude of the cameras with a bundle adjustment algorithm.

Orthorectification requires a model of the camera position and attitude (the extrinsic model), a model of the camera pixel
projections (the intrinsic model), and a model of the Earth's surface (a Digital Elevation Model or DEM).

Our first guess of the orthorectification uses an extrinsic model based on the aircraft's onboard avionics system. The onboard avionics system consists of two identical Honeywell LASEREF III/IV Inertial Reference Units (IRU) and a Garmin Model OEM WASS GPS-16 Unit (GPS) (NSF/NCAR Gulfstream V Investigator's Handbook, https://www.nasa.gov/pdf/594185main_gv_completehandbook.pdf). We use the position from the GPS-corrected IRU longitude and latitude, GPS al-





titude (altitude above mean sea level (MSL) plus the geoid height), and IRU true heading, pitch, and roll. These systems have a stated accuracy of $\pm 15$m in horizontal position, $\pm 10$mfile in vertical position, $\pm 0.05°$ pitch and roll, $\pm 0.2°$ in true heading. Sharp changes in attitude can affect the accuracy of the GPS unit and cause discontinuities in the aircraft position measurements. This was observed during the MethaneAIR flights. Additional attitude errors can result from the alignment and damping of the instrument mounting. As a result, our first guess orthorectification extrinsic model has substantially larger uncertainty

than that reported in the avionics.

We apply a simple intrinsic model using a fixed focal length camera model with no distortion, distributing pixels at equal angles about the boresight ($2 \arctan(f \tan(\theta/2))$), where $f$ is the focal length and $\theta$ is the field-of-view and is $33.7º$). MethaneAIR used Edmund Optics 16 mm Focal Length Lens for 1" Sensor Format (https://www.edmundoptics.com/p/16mm-focal-length-lens-1quot-sensor-format/17861/). The lens has a reported TV distortion of $< 1\%$.

DEM maps were created during processing using the United States Geological Survey (USGS) 3D Elevation Program (3DEP) 1 arc-second data, gathered via the Python `py3dep` package. To cover regions beyond the US for future flights we will replace this dataset by global ALOS DEM at 30 m resolution using our python `demalos` package (https://github.com/ahsouri/demalos)

The orthorectification algorithm proceeds as follows:

– Calculate the aircraft position in meters in Earth-centered, Earth-fixed (ECEF) coordinate system from the longitude,
latitude and elevation, assuming geodetic coordinates on the WGS84 ellipsoid.

– Calculate unit vectors for the tangent frame to the WGS84 ellipsoid (North-East-Down) at the aircraft position.

– Calculate unit vectors of the aircraft frame by applying intrinsic rotations to the tangent frame about the true heading, pitch, and roll.

– Calculate the camera boresight (assumed to be +z in the aircraft frame).

– Calculate individual pixel boresights by applying an additional rotation about the aircraft x axis according to the intrinsic model.

– Calculate reference vectors from each point on the DEM to the aircraft.

– Snap the pixel boresights to the reference vectors (i.e., find the closest reference vector using Euclidean distance in $\mathbf{R}^3$) to find the surface elevation for each pixel projection.

– Calculate the intersection of the pixel projection with an ellipsoid equal to the WGS84 ellipsoid adjusted to the surface elevation.

– Calculate the geodetic longitude and latitude of the intersection of the pixel projections with the Earth.

This algorithm's trade-off between speed and accuracy depends on the resolution of the DEM. When computing the orthorectification for output, we use the DEM with the full 1 arc second. When optimizing the aircraft position and attitude

during bundle adjustment, we aggregate the DEM to 8 arc seconds (applied to the reprojection and the registered image).



We estimate the sun position using the Astronomical Almanac 2020 algorithm and transform from Earth-Centered Inertial (ECI) coordinate to ECEF by propagating through precession, nutation, Greenwich hour angle, and polar motion matrices using Earth orientation parameters from the IERS Bulletin.

A fully automatic geometric correction, which refers to removing geometric distortions in raw images with respect to an or-
thogonal view of a reference image, is developed using a computer vision image registration method. We apply image registra-
tion to the avionics-based MethaneAIR imagery procedure in a two-step process: geographically matching i) the MethaneAIR
$O_2$ imagery to cloud-free orthorectified Multispectral Instrument (MSI) (Sentinel-2) band 11 reflectance ( 1610 nm) at the
top of the atmosphere and ii) $CH_4$ imagery to the rectified MethaneAIR $O_2$ imagery. We name the first (second) step as the
absolute (relative) correction. In the first step, the use of the MethaneAir $O_2$ channel is preferred over the $CH_4$ channel due to
the presence of stronger signal-to-noise ratios. The clear-sky MSI imagery (<3% cloud in each scene covering $100 \times 100$ km$^2$)
at the resolution of 20 m is taken at the nearest time with respect to that of the MethaneAir acquisition enabling to minimize
the potential differences in the surface features changing over time. We reduce the atmospheric interference by averaging the
MethaneAIR radiance within the atmospheric windows of 1250-1255 nm and 1620-1630 nm for $O_2$ and $CH_4$, respectively.
To be able to automatically extract distinctive points (called key-points) from the images, we leverage a multi-scale feature
extraction method called Accelerated KAZE (A-KAZE) (Alcantarilla et al., 2013). This method uses a scale-based non-linear
diffusion filter to describe distinctive features (i.e., corners, lines, patterns, etc.) while reducing noise and simultaneously re-
taining the boundaries of objects. An adaptive histogram equalizer with a window of 8x8 is used to enhance the image contrast,
which is found to be advantageous in increasing the number of key-points. Subsequently, A-KAZE determines extrema by
comparing filtered images by the diffusion filter at different scales (i.e., less to more blurred) in a window of 3×3 pixels. Low
contrast key-points are eliminated by fitting a 2D quadratic function. Each key-point is then described by a 3-bit descriptor
encapsulating radiance intensity and vertical and horizontal gradients pertaining to patches neighboring the key-point based on
Modified-Local Difference Binary (M-LDB) described in Alcantarilla et al. (2013). It is worth noting that each patch is defined
based on the gradient vector, which in turn, makes the algorithm invariant to scale and rotation. We score the Hamming distance
between key-points generated from both MSI and MethaneAIR, remove the irrelevant features based on a fixed threshold, and
use a brute-force matcher to pair them. A robust outlier detector called Random Sample Consensus (RANSAC) is applied to
the paired key-point candidates with more than 10,000 iterations and a residual error threshold of 0.0005°. This algorithm ul-
timately provides a linear relationship between the geographic latitudes and longitudes of the paired key-points extracted from
the target (unrectified) image with respect to those from the reference one. This linear relationship mathematically explains
varying scales and translations between two images, allowing for rectifying major geometric misalignment. This algorithm is
called GeoAKAZE, which is publicly available at (Souri, 2022).

In terms of the absolute correction, we run the geolocation algorithm on the entire set of MethaneAIR flights without sub-
optimally modifying the parameters defined in the algorithm, including the number of the RANSAC iteration and the size
of the histogram equalizer window. The approach was found to be successful, with the results displayed in Table 2. These
numbers are estimated based on assessing individual paired images and considering the residuals obtained from the linear
fit. The relatively poor performance in the RF01 data likely arises from a combination of the aircraft's low altitude and/or





**Table 2.** Statistics from the orthorectification GeoAKAZE procedure for the MethaneAIR (M-AIR) research flights.

| | A-KAZE Statistics | | | Reasons for failures | | |
|---|---|---|---|---|---|---|
| Flight | Total Cases | Success | Failure | Clouds | Bad M-AIR Data | Missing/Irrelevant MSI |
| RF01 | 199 | 135 (68%) | 64 (32%) | 0 (0%) | 42 (66%) | 22 (34%) |
| RF02 | 358 | 314 (88%) | 44 (12%) | 0 (0%) | 3 (7%) | 41 (93%) |
| RF03 | 611 | 523 (85%) | 88 (15%) | 30 (34%) | 22 (25%) | 36 (41%) |
| RF04 | 190 | 164 (86%) | 26 (14%) | 13 (50%) | 4 (15%) | 9 (35%) |
| RF05 | 248 | 240 (97%) | 8 (3%) | 0 (0%) | 0 (0%) | 8 (100%) |
| RF06 | 318 | 301 (95%) | 17 (5%) | 1 (6%) | 7 (41%) | 9 (53%) |
| RF07 | 511 | 491 (96%) | 20 (4%) | 1 (5%) | 7 (35%) | 12 (60%) |
| RF08 | 404 | 381 (94%) | 23 (6%) | 0 (0%) | 7 (30%) | 16 (70%) |
| RF09 | 774 | 578 (75%) | 196 (25%) | 101 (52%) | 43 (22%) | 52 (26%) |

passes over mountainous terrain. Figure 5a depicts some instances of excellent proficiency of the method in orthorectifying the MethaneAIR $O_2$ images to MSI, while Figure 5b shows examples of the respective mapping of $CH_4$ to $O_2$. The background picture for (a) is taken from the MSI band 11 (the reference image), and the overlaid picture is from MethaneAIR $O_2$, while for (b), the background is $O_2$. It is readily evident that features such as roads, farms, and buildings are better lined up after the
corrections.

    The spatial error associated with the avionics-only orthorectification data is depicted in a whisker-plot image; see Figure 6. RF01 has the largest range of errors of any flight, perhaps due to variable flight height and maneuver (Table 2). Except for this flight, the mean value of the median spatial error across the flights is approximately 600 m which is over an order of magnitude greater than the native pixel resolution. This necessitated our need to implement the GeoAKAZE procedure to all flight data.

As for the relative correction, the success rates are found to consistently be over 95% for all the research flights of MethaneaAIR (not shown in Table 2). Since MethaneAIR $O_2$ and $CH_4$ radiances are captured at (nearly) the same time, geometry, and environment, we observe higher success rates compared to the absolute correction, indicating a strong degree of relevancy between the radiances.

    Following this, we attempt to optimize the aircraft avionics (position and attitude of the aircraft) to the GEOAkaze-corrected
pixel projections to produce optimized viewing geometry. The optimized avionics result in values of the viewing zenith angle, scattering angle, geometric airmass factor, etc., consistent with the observed imagery. We use the Limited-memory Broyden-Fletcher-Goldfarb-Shanno algorithm over a bounded domain (Byrd et al., 1995) to derive the corrections to the aircraft position (m ECEF) and attitude (degrees pitch, roll, true heading) that produces the minimum difference between the pixel projections and the GEOAkaze-corrected pixel projections. In cases where the optimized correction falls near the edge of the domain
(initialized to 200m or 10 degrees in each direction), we note the case for quality control checks and expand the domain



and recompute the optimization. We compute the optimization independently for each data granule. This may result in small discontinuities in the orthorectification within the tolerance of the orthorectification error.

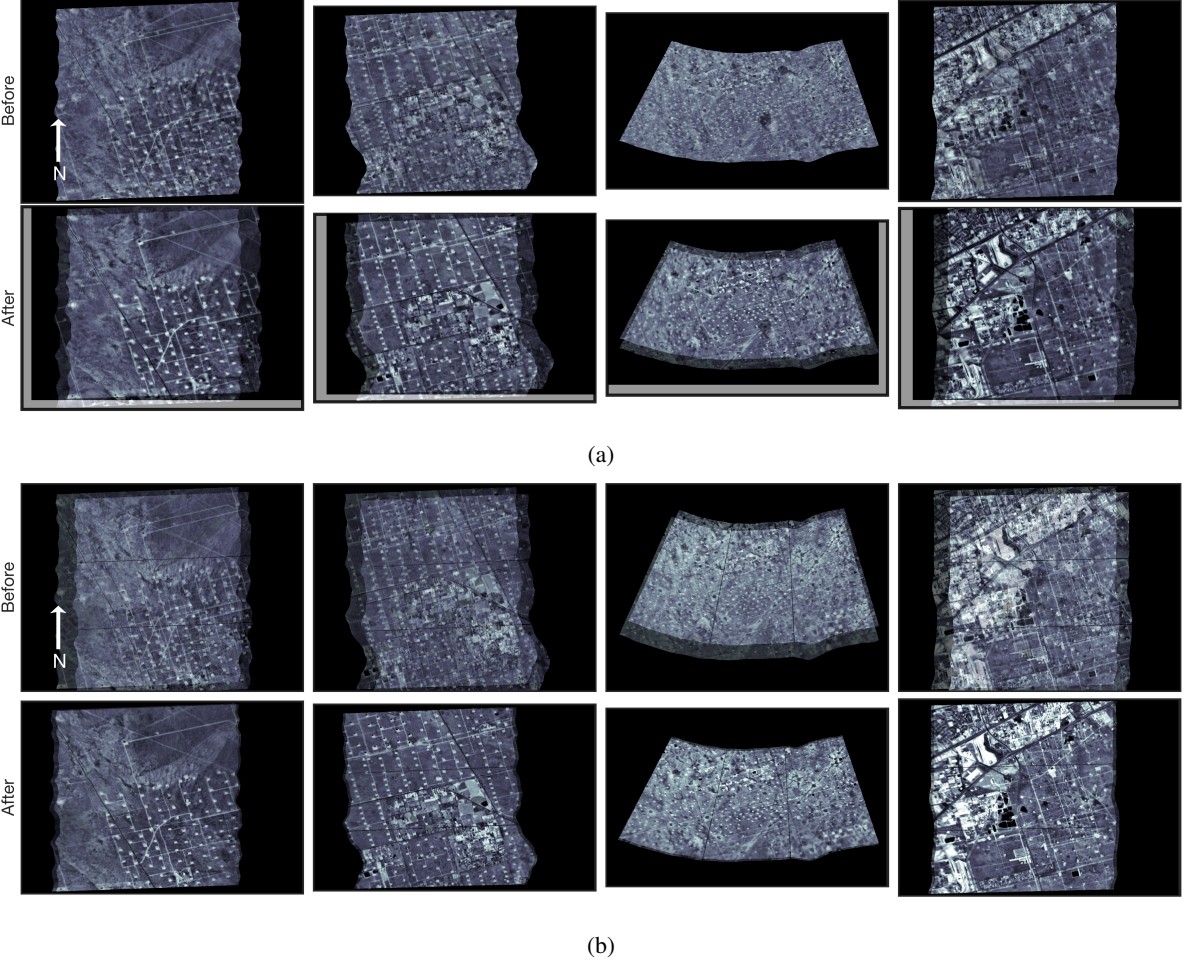

**Figure 5.** Shown in (a) are MSI band 11 imagery (background) overlaid by the spectrally-averaged MethaneAIR $O_2$ channel (foreground) before applying the geolocation correction (first row) and after (second row). In (b), spectrally-averaged MethaneAIR $CH_4$ channel (foreground) overlays spectrally-averaged MethaneAIR $O_2$ channel data (background) before applying the geolocation correction (first row) and after (second row). Note that surface features are markedly aligned between images after the adjustment.

When the GEOAkaze process fails, the orthorectification optimization procedure cannot occur. Therefore, the L1B data utilizes the less accurate avionics-only orthorectification data, which is always available. On the contrary, where the GeoAKAZE is successful, it is still possible for the optimization procedure to fail, either due to the optimizer failing to converge or if the average deviation between the resulting optimized spatial positions and the GeoAKAZE positioning is greater than some defined threshold (which was 25 m for all flights; hence this is the upper bound on the orthorectification optimization accuracy against





the GEOAkaze results). In such cases, we utilize the GeoAKAZE-derived longitude/latitude values in the main L1B Geolocation group together with the avionics viewing geometries. We place the unsuccessful optimized values in a supporting data

group for completeness. In the much more desirable situations where the optimization succeeds, we place the optimized data in the main group and the less accurate avionics data into a supporting group. The results of the orthorectification procedure are, thus, variable; hence, the dashed enclosure applied to the procedure is shown in Figure 2.

As for the failed cases in both absolute and relative geometric corrections, we design a simple and fast empirical method to reduce their geometric offsets as much as possible by leveraging the successful cases from GeoAKAZE. This method finds

a failed segment and nudges it towards the neighbouring successful segments by fitting a linear transformation between their coordinates at the extremities of along-track pixels. The method is carried out iteratively until no further unsuccessful segment exists.

### 3.7 Alignment Between the $O_2$ and $CH_4$ Channels in the Pixel Domain

The observed radiance frames of $O_2$ and $CH_4$ for the MethaneAIR instrument are not aligned in the across–track and along–

track (image) dimensions. This was primarily caused by a boresight offset between two detectors (approximately 25 pixels) and the fact that the sensors were orientated at $180^o$ to each other. While the geolocation and viewing geometry parameters for each granule have been corrected to line up with each band, the offset needs to be determined in the pixel domain so that the ISRF table indices can be well incorporated in the full physics retrieval algorithm where both channels are simultaneously involved.

An example of the offset is presented within Figure 7 (a), for $O_2$ (background) and $CH_4$ (foreground), respectively, where the radiance has been converted to grayscale units. This was done by averaging the radiance data in the wavelength ranges of 1602–1638 nm and 1245–1273 nm for the $CH_4$ and $O_2$ sensors, respectively, normalizing these averages to the maximum observed value in each, scaling by a factor of 255, and then applying the adaptive histogram equalization. Performing a full physics retrieval on such data will not be possible where pixel $[i, j]$ (in terms of along and across dimensions) of the $O_2$ channel

does not align with pixel $[i, j]$ of the respective $CH_4$ granule.

Approximate 30 seconds of L0 data streams are processed into smaller granules of 10 seconds, and in particular instances, it is possible that there is an offset along–track (number of frames) of respective $CH_4$ and $O_2$ granules. The offset along the track is very subtle, often less than 1–2 frames, but across the track the offset is large, evident from Figures 7 (a) and (b).

To deduce the offset corrections, we leveraged the GeoAKAZE relative correction parameters to set a mesh grid of tie points

having identical latitudes and longitudes between two channels. Because the relative correction has co-registered these two bands with respect to each other and this process is irrespective of the performance of the absolute correction, these tie points should be legitimate choices to co-register two images in the image domain. This approach also reduces the computational costs associated with running A-KAZE because it leverages the correction parameters made during the relative correction thus we are no longer required to recall A-KAZE. We identify the $[i, j]$ of the tie points and estimate two offsets, i.e. the offset pixel

numbers in each dimension. In Figure 7 (b), the corrected $CH_4$ gray scale image is shown. For this, an along-track shift of three frames and an across–track shift of 163 were deduced to match two images in the image space. In general, the across–track and



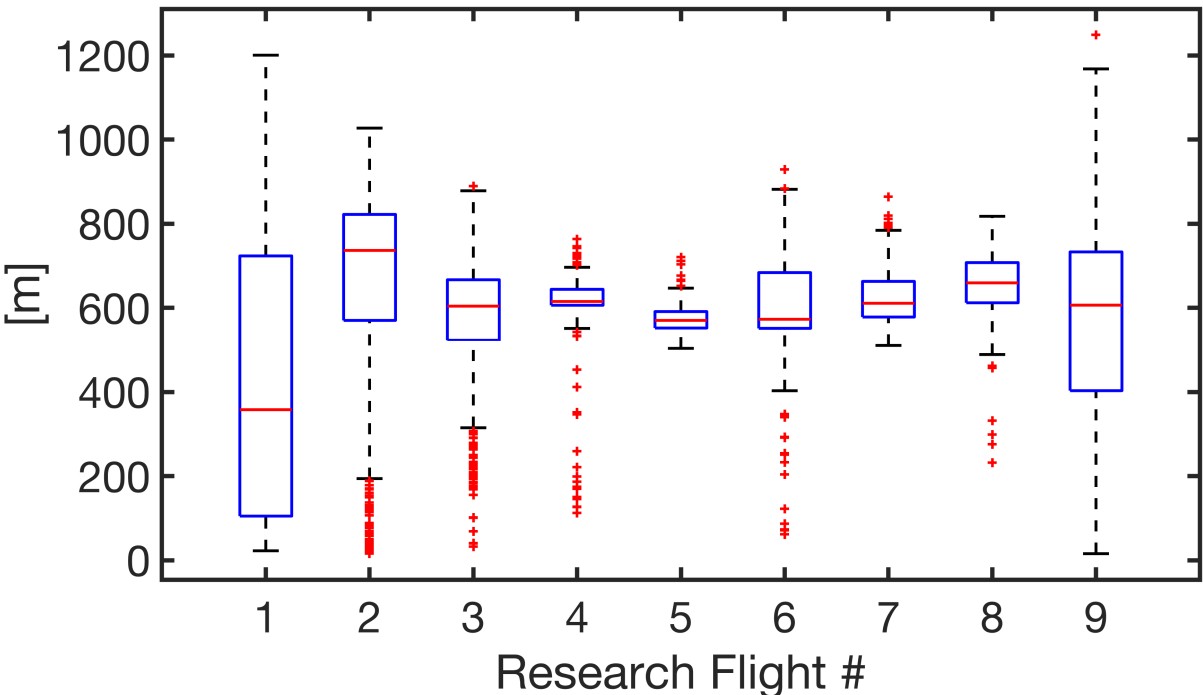

**Figure 6.** The box-whisker plots of geolocation errors (avionics-only orthorectification data) before the corrections. These plots are based on the succeeded cases from GeoAKAZE. On each box, the central red line shows the median, and the top and bottom edges of the box show the 25th (q1) and 75th (q3) percentiles. The dark solid lines at the very beginning and the end of each plot show the minimum and maximum values. Red plus symbols indicate outliers that are determined by any value above $q3 + 1.5 \times (q3 - q1)$ or below $q1 - 1.5 \times (q3 - q1)$.

along–track shifts were found to be $192 \pm 15$pixel and $1 \pm 1$pixel for RF01 and RF02. While the along-track offset remained the same for other flights, the across-track shift was reduced to $159 \pm 5$pixel due to the $O_2$ camera being physically remounted onto the $O_2$ after RF03.

## 3.8 Aggregation

The signal-to-noise ratio (SNR) for individual pixels at the native resolution is approximately 90 and 140 for the $CH_4$ and $O_2$ spectrometers, respectively, which can sometimes be too low for accurately performing retrieval studies. To address this, the native resolution L1B files are supplemented with aggregated L1B data. We aggregated the native data into five segments across–track pixels ($5 \times 1$). By performing this, the SNR is boosted by a factor of 2.2 for a $5 \times 1$ aggregation.





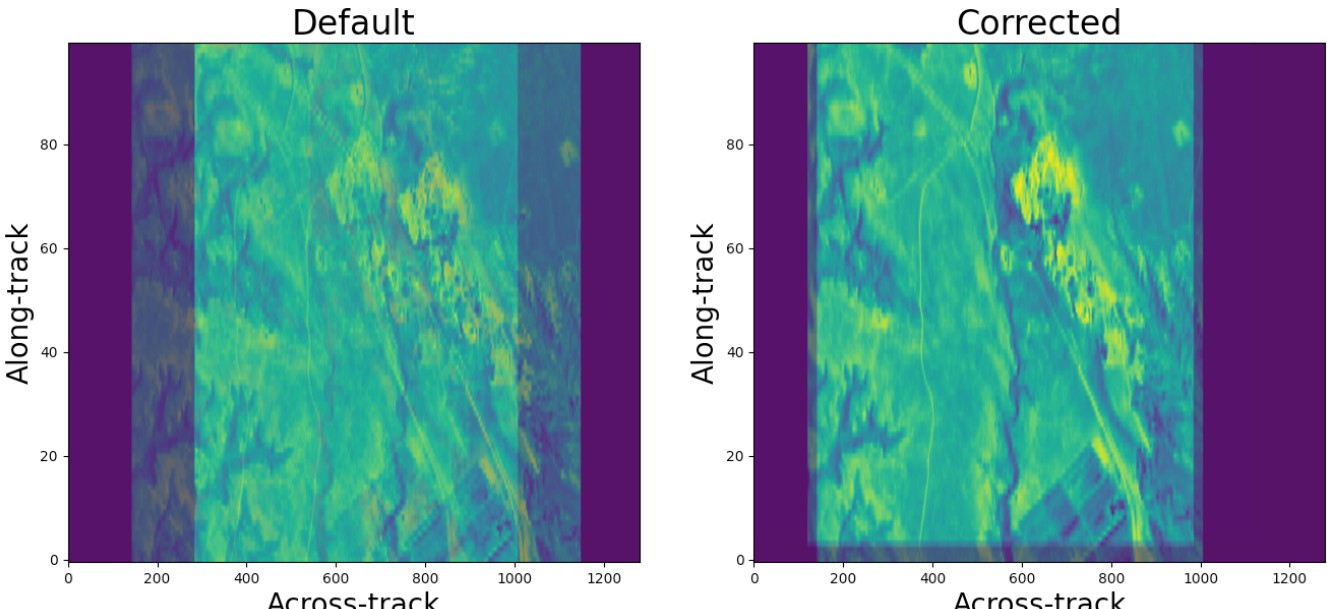

**Figure 7.** An example of the alignment offsets in respectively timestamped $CH_4$ (foreground) and $O_2$ (background) granules of ten seconds in length, procured during RF08. (left) The original radiance images, (right) The overlaid and aligned $CH_4$ and $O_2$ radiance images.

## 4    Conclusion

The mission goal of MethaneSAT is to precisely map over 80% of the production sources of methane emissions from oil and gas fields across the globe to an accuracy of 2–4 ppb on 2 $km^2$ scale. Accomplishing this task requires processing L0 data streams to L1B products with high accuracy. The MethaneAIR airborne simulator has provided critical science data to develop mature and efficient algorithms that will be utilized in the L0–L1B processor for MethaneSAT. Within the observation data of the MethaneAIR research flights, several corrections were required to be developed.

Bad pixels are flagged by considering their standard deviation from the mean dark current, collected at numerous intervals throughout each flight. Stray light was corrected by following the results of Staebell et al. (2021), where a far field kernel is utilized to redistribute light outside of a $15 \times 11$ spectral-spatial window.

A forward model was developed using the latest available spectroscopic data from HITRAN2020. These were combined with a high-resolution solar spectrum and laboratory calibration data for both oxygen and methane sensors. A small increase in cabin temperature which occurred during the second research flight of MethaneAIR caused the instrument line shape to deviate from the laboratory-measured values, which became evident during wavelength calibration. The forward model is demonstrated to accurately capture in-flight modulation of both the wavelength shift and ILS to a reasonable degree of accuracy.

To project MethaneAIR radiance observations onto the Earth's surface (orthorectification), we trace light rays from the instrument, through a camera model, to the surface topography. The orthorectification procedure is subject to errors on the order of 600-800 m, due primarily to uncertainty in the position and attitude of the camera. To correct these errors, we register



MethaneAIR $O_2$ band radiances to existing geo-referenced MSI band 11 imagery using a fully-automated feature extraction method called A-KAZE. We then apply the same algorithm to register MethaneAIR $CH_4$ band radiances to MethaneAIR $O_2$ band radiances. We then optimize the aircraft position and each camera's attitude to best fit the registered data (reprojection).

Surface features in the ortho-rectified images in both flights are well aligned, suggesting significant improvements in the geolocation accuracy for most scenes (>90%). Nonetheless, we are aware of two important limitations of this approach which are i) the presence of cloud and ii) a long lag between the acquisition time of MSI and MethaneAIR for some cases due to our stringent cloud screening cut-off value resulting in unmatchable key-points. This automatic algorithm, named GeoAKAZE is publicly available (Souri, 2022) and can be used for other airborne missions.

The misalignment of the radiance data recorded by the methane and oxygen sensors initially makes them unsuitable to be used in the full physics retrieval algorithms, which requires the use of both channels. By converting the radiance data to grey-scale images, we demonstrate via the use of the relative correction parameters made by A-KAZE algorithm that the offsets (in pixel space) between the two sensors can be deduced to high accuracy and reliability. In this approach, the methane data were aligned to the oxygen sensor's data, and frames were added or subtracted to create a full matching set of data when necessary.

This misalignment is expected to be much smaller for MethaneSAT.

The MethaneSAT L0–L1B processor will include all the algorithms described in this article, although not all may be required, and some may need to be modified.

*Code and data availability.* The L0-L1B processor and data are available upon request.

*Financial support.* This research was funded by MethaneSAT LLC and an NSF EAGER grant.

*Author contributions.* EC, AS, JB, KS, CS, BL developed scientific algorithms. JW, SR, CCM and AC contributed to algorithm design. IG contributed towards the spectroscopic data. JS, MS, JH and BD provided expertise on instrument design. XL, JF, KL and SW provided oversight on the campaign flight plans, algorithm designs and overall project guidance.

*Competing interests.* The authors declare that they have no conflict of interest.



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
