# Peer review of "Level0-to-Level1B processor for MethaneAIR"

_Atmospheric Measurement Techniques, 2023_

## Referee Comment (RC2)

Review of AMT-2023-111

Level0-to-Level1B processor for MethaneAIR

By Eamon K. Conway, Amir H. Souri, Joshua Benmergui, Kang Sun, Xiong Liu, Carly Staebell, Christopher Chan Miller, Jonathan Franklin, Jenna Samra, Jonas Wilzewski, Sebastien Roche, Bingkun Luo, Apisada Chulakadabba, Maryann Sargent, Jacob Hohl, Bruce Daube, Iouli Gordon, Kelly Chance, and Steven Wofsy

**General comments.**

  The authors present a processing chain from Level 0 (raw observation data) to Level 1B (radiometric calibrated and georeferenced data) for MethaneAIR data. I understand that this activity is important to retrieve a good $CH_4$ products, which will use for identifying the oil and gas filed hot spot, from MethaneAIR data. However, some description is unclear or missing in the text. Unfortunately, it is hard to follow the processing chain. So, I recommend the authors will add the sentences and clarify for some of unclear sentences.

  For these reasons, I recommend this paper for publication with major changes to the technical content.

**Specific comments.**

**Affiliation**

1.  Page 1:  Change from "Center for Astrophysics | Harvard and Smithsonian" to "Harvard - Smithsonian Center for Astrophysics," and remove ".02138".

**Abstract**

2.  Page 1, line 1: Spell out "L0" and "L1B".

3.  Page 1, line 4: Spell out "EDF".

4.  Page 1, line 5: The accuracy is defined as concentration in unit of ppb. In the nominal case, emissions are expressed as flux. Then, the readers have incongruous with emission accuracy in unit of ppb. Recommend the rewording.

5.  Page 1, line 8: Spell out "ISRF"

**1.  Introduction**

6. Page 2, line 35: Same comment as comment #4.

7. Page 2, line 45: Correct "Greenhouse Gas …" to "Greenhouse Gases …".

**2. The MethaneAIR Instrument**

8. Page 4, Figure 1: The flight route for RF10 is not presented in Fig.1. However, in table 1, the flight duration is presented as 4.7 hrs. It is unclear that RF10 is actual flight or on-ground activity. If RF10 is still on flight activity, the fight route has to be indicated in Fig.1.

**3. L0-L1B Development**

9. Page 5, line 4: The processing time is 180 times longer than that of actual observation duration. Then, it seems that the selected scene data is only processed with this proposed processing chain. In this case, the authors also have to touch on the scene selection process before Level 0 to Level 1B processing as well as some limitation such as duration for one selected scene.

10. Page 5, line 159: How to determine the "electronic offset" value?

11. Page 5, Figure 2: How to handle the aircraft window transmission in the processing chain? The processing box is presented in Fig.2.   However, it is no explanation in the text. The authors have to touch on this topic.

12. Page 7, Eq.2: What is the meaning of $\lambda'$, $\delta\lambda$, $Al_k$ and $B_j$ ?

13. Page 9, Figure 3: What is the required output spectral range for MethaneAIR $CH_4$ retrieval? In the text at line 39, the spectral range from 1598 nm and 1683 nm is defined for observation. However, the plot for $CH_4$ is started around 1602 nm and ended around 1655 nm. The authors have to be clearly described the output spectral range for MethanAIR L1B file both $CH_4$ and $CO_2$ bands.

14. Page 11, Figure 4: It is hard to distinguish between "Gaussian Filtered ISRF" and "LUT ISRF", due to the same color. Update to clearer plot with changing the color or the size of symbol.

15. Page 12 line 285: The onboard avionics are mounted on the aircraft and these data are used for aircraft operation. The mounting geometry between the aircraft and instrument platform is not taken account in the first guess. Then, the systematic

error source for line of slight (LOS) is the 3D-rotation of instrumental platform against the aircraft base. Is this understanding correct? Usually, including the satellite sensors, the position and rotation angle of optical bench during observation is monitored and applied for LOS processing. Why the additional IRU unit is not mounted on the optical bench? It seems that the processing cost for first guess will be reduced by applying onboard IRU data.

16. Page 13, line 313: Spell out "IERS".

17. Page 13, line 319 and 321: Typo "MethanAir"

18. Page 17, line 407: The across-track offset is estimated as 150 to 200 pixels (rows). $O_2$ and $CH_4$ camera have 1280 pixels in rows. In this case, co-aligned pixels are almost 1000 pixels in rows. In the text, L1B files which indicated in Fig.2 is not described and it is unclear that all the output information of this processing chain. The authors have to describe the output (L1B file) of this processing chain. It seems that the effective across-track information is around 1000 location instead of 1280.

**Reference**

19. Page 22, line 554, Staebell et al., now published in Atmospheric Measurement Techniques. The authors have to be updated the reference.

End of document

---

## Author Comment (AC1)

**Response to Reviewer #1.**

We would like to thank the reviewer for taking the time to review our article, and for providing critical feedback on the manuscript. Below, we cycle through each of the comments in order of their appearance and provide a response. All edits will be implemented into the final manuscript where necessary.

*(1) The CH$_4$ measurement with accuracy of "2-4 ppb on 2 km$^2$" is very challenging. The accuracy by remote sensing technique with existing spectrometers using solar reflected light is about 10 ppb. The L1B product is usually defined as calibrated radiance spectra. What are required for characterization, calibration and level 1 products? How stable the wavelength or the instrument response function of the spectrometer should be during the flight to achieve accuracy of 2-4 ppb? How accurate are radiometrically calibrated (in Figure 2)?*

**This is certainly a challenging question to answer and will require a dedicated description of the MethaneSAT platform, and a future publication is planned for MethaneSAT that will address these requirements. However, for the present paper, for MethaneSAT, our models predict that an instrument SNR of approximately 1,000 will provide the necessary requirements to retrieve CH4 to a precision of 2-4 ppb over 2km2. We have added this to the document to provide a general idea of the instrument specifications of MethaneSAT.**

*(2) Chapters. The title of the Chapter 2 is "instrument", but it describes the research flight. There is only "2.1" and no "2.2".*
**We have altered the numbering of this section, to follow Chapter 2: MethaneAIR, 2.1: MethaneAIR Instrument, 2.2: Research Flight Details.**

*(1) Page 4, Line 87 "Are not monitored in the L0-L1B"*
*Does it mean "Laboratory-measured wavelength is not used. Wavelength is tuned during the trace gas retrieved"?*
**For TROPOMI L0-L1B processing, they did not implement an algorithm to monitor for changes in the wavelength, the lab measurements were not optimized. They use the lab-measured data only. Indeed, the wavelength is fine-tuned/optimized in the L1-2 algorithm for the retrieval. We adjusted the sentence.**

*(2) Page 4, Figure 1*
*The description of the area such as Colorado, New Mexico etc. and legend of "RF **" will help readers' understanding.*
**We have added state names to the image such that one can track the flight path over the various U.S. states.**

*(3) Page 10, Line 258, "approximately 30 %". Does it mean "difference in FWHM of ILS"?*
**Yes, it does – we have clarified the text to make this clear to the readers.**

*(4) Page 10, Line 264, Page "in order to pinpoint the exact location"*
*How accurate the orthorectification should be from the cruising altitude?*
**The accuracy of the orthorectification will determine the positioning accuracy of the point sources and is proportional to the sampling power of the instrument. GEO-AKAZE provides positioning accuracy that is less than approximately 30 m, the resolution of the base MSI imagery. We have added a sentence.**

*(5) Page 11, Figure 4*
*It is difficult to distinguish between bold dots and asterisks.*
*There are more than one plots at indices of 500 and 1000. Why?*
**We have made the points (dots and stars) larger to make it clear there are two different sets of data being plotted here: Lab measured ISRF FWHM and calculated. The colors represent the various ISRFs – we have added more context to the caption. The 500/1000 erroneous points are part of the same set of ISRF lab measurements.**

*(6) Page 18, Line 425 "small increase in cabin temperature"*
*It is not clear how small the increase is. The actual temperature variation will help readers understanding.*

**We have specified that the temperature increase was very limited, 3-4 degrees Kelvin.**

*(1) Page 4, Line 90, "for each sensor"*
*Does it mean $CH_4$ and $O_2$ spectrometers?*
**Yes, we have adjusted the text.**
(2) Page 12, Line 281, "+/- 10 m file". Is it "+/- 10 m"?
**Yes, this is a typo and has been adjusted.**

---

## Author Comment (AC2)

**Response to Reviewer #2.**

We would like to thank the reviewer for taking the time to review our article, and for providing critical feedback on the manuscript. Below, we cycle through each of the comments in order of their appearance and provide a response. All edits have been implemented in the revised manuscript, where necessary.

**Affiliation**

1. *Page 1: Change from "Center for Astrophysics | Harvard and Smithsonian" to "Harvard - Smithsonian Center for Astrophysics," and remove ".02138".*
   a. **Some years ago (1-3) the CfA changed its name from** *"Harvard - Smithsonian Center for Astrophysics,"*, **to** *"Center for Astrophysics | Harvard and Smithsonian"*. **02138 will be removed.**

**Abstract**

1. *Page 1, line 1: Spell out "L0" and "L1B".*
   a. **We have spelled out 'L0' and 'L1B'.**
2. *Page 1, line 4: Spell out "EDF".*
   a. **We have expanded 'EDF'.**
3. *Page 1, line 5: The accuracy is defined as concentration in unit of ppb. In the nominal case, emissions are expressed as flux. Then, the readers have incongruous with emission accuracy in unit of ppb. Recommend the rewording.*
   a. **We apologize for the incorrect usage of 'emissions' in this context. We have removed 'emissions' from the sentence.**
4. *Page 1, line 8: Spell out "ISRF"*
   a. **We have expanded ISRF.**

**Introduction**

1. *Page 2, line 35: Same comment as comment #4.*
   a. **We apologize for the incorrect usage of 'emissions' in this context. We have removed 'emissions' from the sentence.**
2. *Page 2, line 45: Correct "Greenhouse Gas …" to "Greenhouse Gases …".*
   a. **We have changed 'gas' to 'gases'.**

**The MethaneAIR Instrument**

1. *Page 4, Figure 1: The flight route for RF10 is not presented in Fig.1. However, in table 1, the flight duration is presented as 4.7 hrs. It is unclear that RF10 is actual flight or on-ground activity. If RF10 is still on flight activity, the fight route has to be indicated in Fig.1.*
   a. **We have added RF10. We omitted it as it was not measuring CH4 emissions, but rather for recording airglow.**

**L0-L1B Development**

1. *Page 5, line 4: The processing time is 180 times longer than that of actual observation duration. Then, it seems that the selected scene data is only processed with this proposed processing chain. In this case, the authors also have to touch on the scene selection process before Level 0 to Level 1B processing as well as some limitation such as duration for one selected scene.*

a.   The data for an entire flight is always processed in this chain. We should have clarified that each L0 granule that is collected covers 30-seconds of observation.

2. *Page 5, line 159: How to determine the "electronic offset" value?*

   a.   **The electronic offset was determined during instrument calibration [Staebell et al.]. It is the dark current at zero integration time. It is '1500' DN. This has been added to the document.**

3. *Page 5, Figure 2: How to handle the aircraft window transmission in the processing chain? The processing box is presented in Fig.2. However, it is no explanation in the text. The authors have to touch on this topic.*

   a.   **We have added a few sentences that discusses the window transmission in section 3.5 – 'Wavelength and ILS Calibration'.**

4. *Page 7, Eq.2: What is the meaning of $\lambda'$, $\delta\lambda$, $Al_k$ and $B''$ ?*

   a.   **The integration variable (lambda prime) spans the wavelength range of the ILS convolution. Delta lambda represents the wavelength shift. Al_{k} represents variables that are optimized in the LSQ fit, and scale the absorption factors (A, B, C). B_{j} are also numerical values that are optimized and shift the observation. These explanations have been added.**

5. *Page 9, Figure 3: What is the required output spectral range for MethaneAIR CH4 retrieval? In the text at line 39, the spectral range from 1598 nm and 1683 nm is defined for observation. However, the plot for CH4 is started around 1602 nm and ended around 1655 nm. The authors have to be clearly described the output spectral range for MethanAIR L1B file both CH4 and CO2 bands.*

   a.   **The retrieval window range will not be the same as the window range considered here. This window is to derive the wavelength shift that is occurring, not to retrieve CH4. The QE of the instrument decreases towards the ends of the spectra and can increase the noise in the fit, hence the ends were omitted. We have added an explanation to the text.**

6. *Page 11, Figure 4: It is hard to distinguish between "Gaussian Filtered ISRF" and "LUT ISRF", due to the same color. Update to clearer plot with changing the color or the size of symbol.*

   a.   **The colors are meant to be the same between the two different models, but the shape of the points are 'stars' and 'dots' – we have made the data points larger to enhance the distinction.**

7. *Page 12 line 285: The onboard avionics are mounted on the aircraft and these data are used for aircraft operation. The mounting geometry between the aircraft and instrument platform is not taken account in the first guess. Then, the systematic error source for line of slight (LOS) is the 3D-rotation of instrumental platform against the aircraft base. Is this understanding correct? Usually, including the satellite sensors, the position and rotation angle of optical bench during observation is monitored and applied for LOS processing. Why the additional IRU unit is not mounted on the optical bench? It seems that the processing cost for first guess will be reduced by applying onboard IRU data.*

   a.   **This analysis is correct, there is a 3-D rotation between the instrument platform and the aircraft base, which is where we receive the initial parameters of the instrument orientation and position in 3-D space. We have not performed an analysis of the numerical transformation variables that the orthorectification is calculating for all observation data files, but this constant 3-D rotation should be common to all flight observations. We have added a line to the document (conclusion) that suggests we will take this into account in future editions of the L0-L1B processing chain to increase the efficiency.**

8. *Page 13, line 313: Spell out "IERS".*
   a. **IERS has been spelled out.**
9. *Page 13, line 319 and 321: Typo "MethanAir"*
   a. **Typo corrected.**
10. *Page 17, line 407: The across-track offset is estimated as 150 to 200 pixels (rows). O2 and CH4 camera have 1280 pixels in rows. In this case, co-aligned pixels are almost 1000 pixels in rows. In the text, L1B files which indicated in Fig.2 is not described and it is unclear that all the output information of this processing chain. The authors have to describe the output (L1B file) of this processing chain. It seems that the effective across-track information is around 1000 location instead of 1280.*
    a. **We have added a line to the document that explains that the effective number of pixels in the across-track dimension is in fact not the full 1,280 that the sensor possesses, but rather, it is approximately 950 due to how light falls onto the sensor.**

**References**
1. *Page 22, line 554, Staebell et al., now published in Atmospheric Measurement Techniques. The authors have to be updated the reference*
   a. **We have adjusted this reference.**

---

## Author Response (AR2)

**Response to Handling Editor.**

We would like to thank the editor for taking the time to review our article, and for providing critical feedback on the manuscript. Below, we cycle through each of the comments in order of their appearance and provide a response. All edits have been implemented in the revised manuscript, where necessary.

Comments
1. L181-182. Summarize some of the "various reasons" for straylight for the reader's benefit.
   a. **We have added several examples of why they may be encountered.**
2. L185. What do you mean by "stray-light kernels"?
   a. **The stray light kernel is used to correct for unwanted (stray) light falling onto a region of the FPA. The kernel extends pixels in the spectral and spatial domain and corrects the contribution of light to a pixel's signal.**
3. L201. When you say "the parameterization of lambda with respect to spectral pixel index is assumed throughout" do you mean that lambda is the wavelength corresponding to the i-th spectral pixel but you omit the i subscript in the equation?
   a. **Yes, that is correct.**
4. Eq. 2. Are Doppler shifts due to Sun-Earth relative motion taken into account in computing the high-resolution solar spectrum? Or are they considered irrelevant for this purpose? I am asking because it is not uncommon that they are accounted for in GHG remote sensing from high-resolution SWIR spectrometers (e.g., O'Dell et al., 2012; Connor et al., 2016).
   a. **Yes. The solar spectrum includes this effect. However, the spectral resolution of the instruments is likely too large for this effect to be noticeable.**
5. L305-309. A lot of prior knowledge is required from the reader to understand references to ECEF, ENU reference frames, etc., but I suspect not many readers of this journal have such knowledge. It may be useful to explain these reference frames in an appendix or provide a reference to where the reader can look them up.
   a. **We have provided an extra few words prior to describing the orthorectification steps that will aid an interested reader to understand the process on a deeper level.**
6. L344. If I understand correctly, A-KAZE looks for the same features in the MethaneAir and in the MSI images. Is the Hamming distance used to assess which feature in one image corresponds to which feature in the other image? This was not entirely clear to me.
   a. **Yes. We have clarified this in the article text.**
7. L346. Add a citation to a paper describing the RANSAC method.
   a. **Yes, this has been added.**
8. Table 2. It is not exactly clear to me how you define a "success" and how you define a "failure".
   a. **A thorough description of how the success/failures are determined has been added to the document.**
9. L365. "Necessitated our need to..." -> "necessitated us to"?
   a. **Corrected in the document.**